# Experimental Investigation on Inner- and Inter-Strip Reinforcements for 3D Printed Concrete via Automatic Staple Inserting Technique

Xiangpeng Cao , Shiheng Yu and Hongzhi Cui *

Key Laboratory for Resilient Infrastructures of Coastal Cities (MOE), College of Civil and Transportation Engineering, Shenzhen University, Shenzhen 518060, China; cxp2007@outlook.com (X.C.); yush199805@outlook.com (S.Y.)

* Correspondence: h.z.cui@szu.edu.cn; Tel.: +86-755-26917849

**Abstract:** Lack of reinforcements is an existing drawback of 3D printed cementitious components, which is an urgent concern. A staple-inserting apparatus was developed and installed on a 3D printer and automatically fabricated 3D printed and staple-reinforced components with 98% successful insertion to achieve inner- and inter-reinforcement of the printed strips. The inserted staples inside the printed strips improved the compressive strength by 25% maximum owing to the inner locking effect by the staple pins, while the flexural strength did not increase because the scattered staples functioned separately. The staples over the strip interfaces remarkably increased the flexural stress by 46–120%. The inserted staples demonstrated a significant strip locking effect, but the unavoidable voids decreased the bonding between staples and the composite. The mechanical analysis concluded that the printing parameters considerably affected the reinforcing rate. The staple inserting technique proved the feasibility of automatic fabrication of fiber-reinforced and printed concrete structures.

**Keywords:** 3D concrete printing; staple reinforcement; auto staple inserting; staple-inserting technique; staple locking effect



## 1. Introduction

Three-dimensional printing technology with digital, customized, and new material features is a promising emerging technology. After more than 20 years of development, 3D technology has been used in aerospace, medical equipment [1,2], and metal parts [3], owing to its high accuracy, short production cycle, and diverse materials. The application and research of 3D concrete printing (3DCP) have gradually increased [4], from Pegna using steam injection to cement powder in 1997 [5], to Contour Craft technology [6] printing curved walls in 2003, and Shanghai Winsun Ltd. Printing various types of houses and many printing applications worldwide in recent years.

Among existing digital concrete fabrication methods, extrusion-based 3DCP is the most widely used and developed technique. Besides the studies of shape retention and the mix formulation [7–9], the reinforcements play an important role in 3D printed components to meet structural safety requirements. The printed output has low toughness due to the layerwise fabrication process and lack of reinforcements. There are layers and filaments from the printed nozzle, stacking, and bonding to form a 3D object, between which there are interfaces during the stacking process of the 3DCP, and the interfaces are the weakness zone [10–12] or, worse contact [13] area, requiring reinforcement. Several studies have been conducted to improve the layer adhesion by controlling the moisture, varying the printing speed and layer thickness, or brushing black carbon–sulfur or concrete glue.

Increasingly, researchers have attempted to introduce fibers/rebar to reinforce printed components [14–17] along with the printing procedure. Studies on reinforcing 3D-printed concrete have been carried out using fibers, wires, cables, chains, meshes, nails, rebar, or

rods. These reinforcements could be applied inner, between, or outer filaments/layers, as demonstrated in Figure 1. Inner reinforcement means that the reinforcements stay only inside the extruded strips. Short reinforcements can mix with the cement powders at the dry stage, transmitted to the printing head, and then extruded out of the printing nozzle within the mixed mortar. These short fibers can be steel fibers [16,18,19], polyvinyl alcohol fiber [18], carbon fiber [17], bio-based fibers [19,20], polymeric fibers [14,16], basalt fibers [17], or glass fibers [21]. The short fibers with a high percentage [22] considerably reinforced the printed output. The compressive strengths of 3D-printed to fiber-reinforced concrete can be up to 107 Mpa [23]. Long reinforcements, such as steel chains [24], steel cables [13], fish lines [25], and wires [25–28], can be embedded into extruded filaments during the printing process. The reinforcements can also be placed across the interfaces to lock the adjacent layers by inserting steel rebar [29,30], nails [31], metal-printed rebar [16], or plastic-printed rebar [16]. Nails inserted into the layered composite improved the bending force by 50%. Wire meshes [32] and rebar [33], placed between the layers, were also considered across the interface reinforcement. The U nails [34] (6 mm in width and 22 mm in height) considerably improved the ultimate tensile strength and shear strength of 3D-printed concrete by 145.0% and 220.0%, respectively. There are still reinforcements outside the printed parts. Steel rods can be installed on the printed and cured components to enhance structural toughness [35]. Printed mesh [36], steel wire meshes [37,38], and steel cages can be placed in the printed contour before mortar grouting.

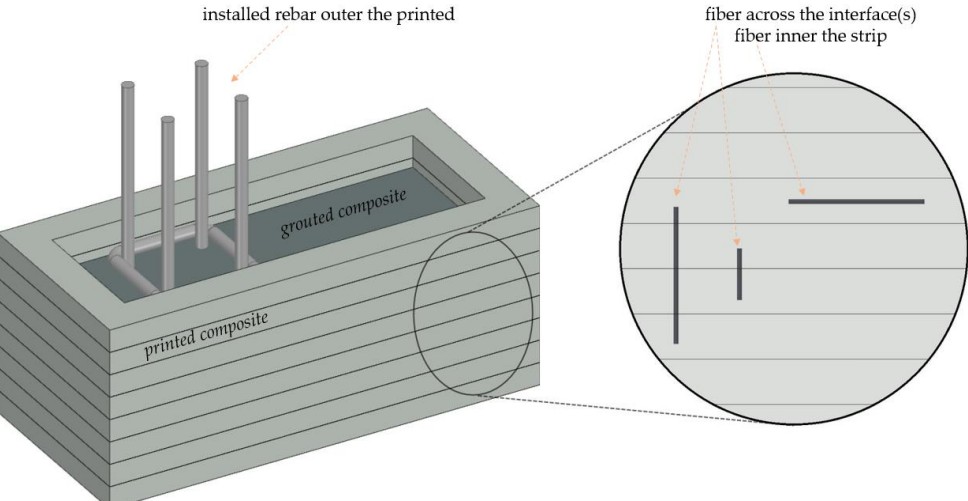

**Figure 1.** Locations of the reinforcements in the printed structure: inner strips, across the interfaces, and outer printed.

However, the previous techniques still have challenges, especially in engineering applications. Short reinforcements require thin and soft fibers to avoid blocking the extrusion; large and long reinforcements cannot be easily manipulated automatically. Cables need low stiffness to be easily redirected as the printing direction turns; otherwise, soft cables easily become knotted [13], thereby increasing the difficulty of automatic embedding. The embedding operation requires a feeding apparatus to synchronize the cables/wires with the printing nozzle motion to maintain less displacement between the fiber and composite. The embedded long reinforcements could be dragged or slacked if not synced; thereby, significant gaps occur within the printing results. The insertion and placement require stable operation. The operating deviation increased the hole diameter in the composite [39], as it could not be avoided from swaying in the insertion and led to voids between the rebar and the composite mixture. The bonding stress depends on the contact area, which varies with the placement angle of the rebar and printing direction, owing to the existing voids [33]. Therefore, the short fiber inserting technique is the most promising solution

among all, since the fiber and its insertion are separate. However, few studies of automatic fiber insertion have been conducted.

In this study, we fabricated all the printed–inserted fiber-reinforced samples automatically, taking the staples as the short steel fibers. Staples commonly fasten wood pieces in packaging industries with complete stapling and feeding mechanisms, which can be migrated into the 3DCP technique as steel fibers with end hooks. Each staple or its insertion is separate and does not affect other staples, then the abovementioned voids and stretches can be minimized with the automatic printing–inserting operations.

To study the effectiveness of the reinforcing technique in actual automatic fabrication, we initially developed the staple inserting mechanism and attached it to a 3D concrete printer. Then, we designed and fabricated the printing–inserting samples automatically to investigate the staple locking effect for the inner- and inter-printed strips. This study provides a solid foundation for future research and development on steel fiber concrete with digital and automatic fabrication.

## 2. Materials and Devices

### 2.1. Printable Material

The printable cementitious mixture used herein was developed in our laboratory, as listed in Table 1, with the ordinary Portland cement (OPC 42.5 R, purchased from Foshan, China) as the basic material. Glass beads (GB; mean particle size of 60 to 80 μm and density of 0.60 g/cm$^3$) were used as fine aggregates to increase the extrusion efficiency owing to their smooth appearance. Admixtures of fly ash (FA; fineness of 43 μm and density of 2.7 g/cm$^3$, the specific surface area of 0.36 m$^2$/g, and 0.5% water content) and silica fume (SF; density of 2.2 g/m$^3$, the average particle size of 0.1–0.3 μm, and the specific surface area of 20–28 m$^2$/g) were chosen to improve the workability of the mortar. High-range water-reducing agents (HRWR; polycarboxylic acid superplasticizer) and thickening water-retaining agents (HPMC; hydroxypropyl methylcellulose) were chosen to improve the printability of the paste by improving its viscosity [40,41]. A plasticizer (Plastic Agent, PA) was used to improve the plasticity of the fresh paste. An accelerator (AR; lithium carbonate) helped to accelerate the hydration. Thixotropic agent (TA): magnesium aluminum silicate (MgAl$_2$(SiO$_3$)$_4$) was chosen to increase the fluidity of the paste during extrusion. In addition, a defoaming agent (DFA) eliminated the air bubbles generated during the mixing and extrusion processes.

**Table 1.** Mix proportion of the 3D-printable material (unit: gram by weight).

| OPC | GB | FA | SF | Water | HRWR | HPMC | PA | AR | TA | DFA |
|------|------|------|------|-------|------|------|------|------|------|------|
| 2000 | 400 | 294 | 106 | 800 | 4 | 6 | 20 | 15 | 30 | 1 |

### 2.2. Printing Device

A frame-type 3D printer was designed and built using an open-source 3D printing control motherboard with a mechanical accuracy of 0.1 mm for the three alignments of the X, Y, and Z axes. The printer has a maximum movement speed of 100 mm/s in the XY plane, with a maximum printing size of 60 × 60 × 100 cm. The resolution of the motion system was 0.1 mm, which is acceptable for concrete printing operations. The control unit of the printer ensured precise motion positioning and stable execution for a smooth printing process. The extrusion bin was fed locally by shovel. The developed extruder with a two-segment screw is shown in Figure 2c. The large screw blade fed the mortar to the small screw, and the small screw then extruded the mortar out of the nozzle tip. A rectangular alternative nozzle tip of 40 × 10 mm, as shown in Figure 2b, was utilized to print a specimen of dimensions 40 × 40 × 200 mm.

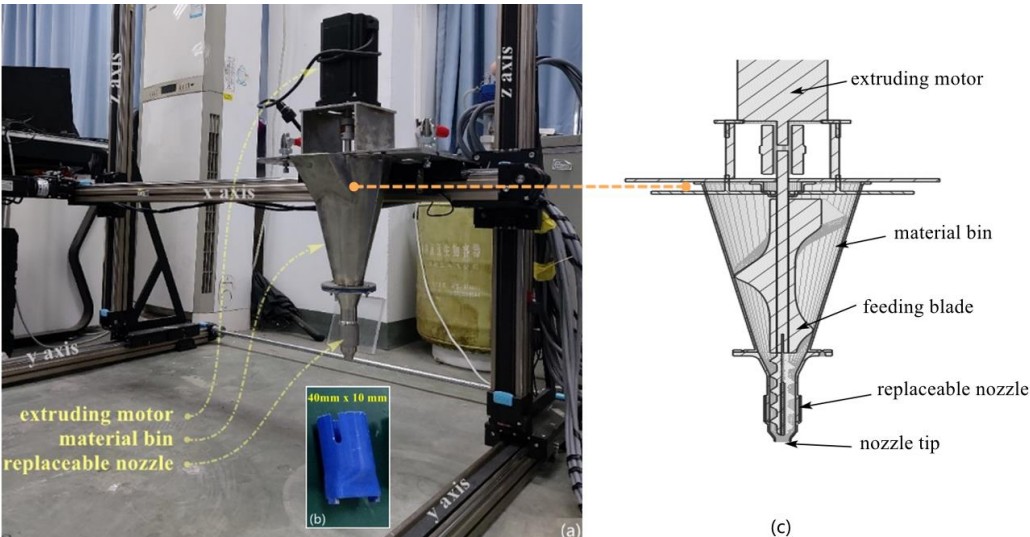

**Figure 2.** Three-dimensional printing device: (**a**) 3D printer with a round nozzle tip, (**b**) the rectangular nozzle tip of 40 × 10 mm, and (**c**) design of the printing head.

### 2.3. Staples and Inserting Apparatus

The staples (type 1008 in this study) were purchased from the industrial market, with detailed information in Figure 3a,b. Each staple had a weight of 0.11 g and a volume of 19.5 mm$^3$. The direct tensile strength was obtained as 642 Mpa, with maximum tensile stress of 450 N for one staple. Staples are commonly used with a stapler, which has a strong spring as the insertion power source, as shown in Figure 3c. It could release the force rapidly and jet the staple into the cement at high speed, but controlling it automatically was challenging. We then altered the original spring with a stepper motor and a screw rod, which the digital signal pulses can control, namely the staple inserting apparatus (SIA). The SIA maintains a low but controllable inserting speed to penetrate the staples into fresh cement, as shown in Figure 3e, with a short signal pulse (voltage from low to high) from the controller to strike a staple.

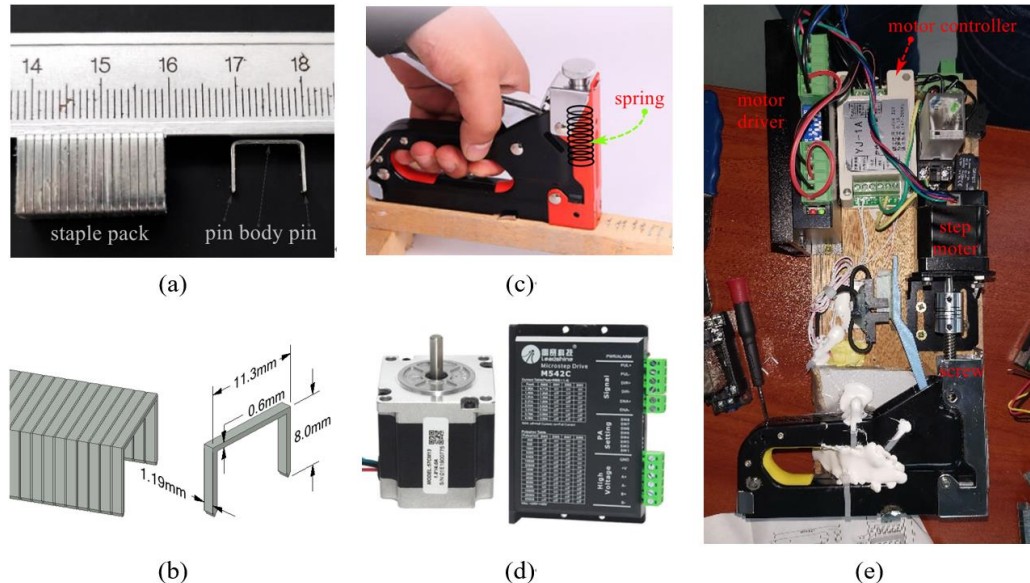

**Figure 3.** Staples and the automatic staple-inserting device: (**a**) photo of staples, (**b**) dimensions of the staple, (**c**) original stapler for wood fastening, (**d**) a step motor and its driver, and (**e**) the upgraded SIA.

### 2.4. The 3DCP-SIT System

The SIA was installed on the existing 3D printing device. The 3D concrete printing and the staple inserting technique (3DCP-SIT) cooperate as one system to fulfill the automatic stapling process, as shown in Figure 4a. The printer control board sent a low-voltage command (M106) and a high-voltage command (M107) to the SIA to strike one staple into the printed mortar, and the inserting commands were merged with the printing commands, as shown in Figure 4b. After one layer was printed, the SIA inserted the staples one by one. The staple-reinforced structure was eventually fabricated layerwise. The automatic 3DCP-SIT system guaranteed that the inserted staple arrays maintained good consistency; as a result, the reinforcement study was scientific.

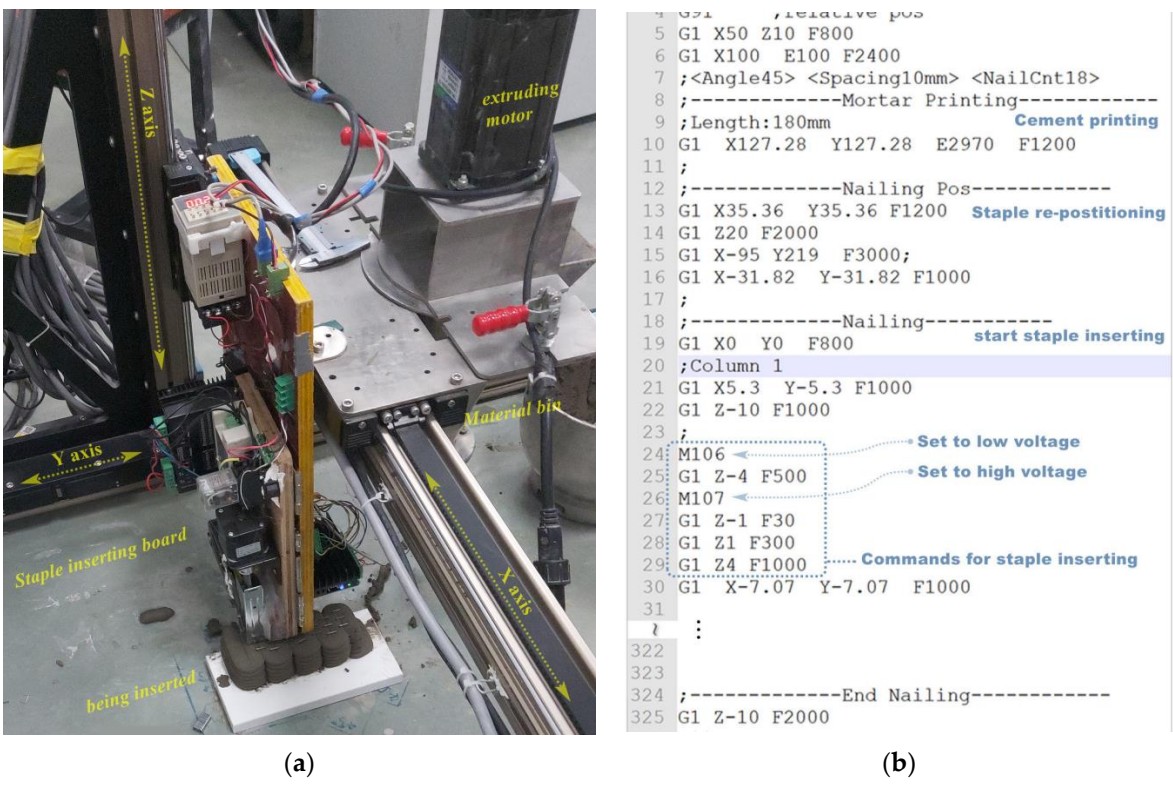

(**a**)　　　　　　　　　　　　　　　　　　　(**b**)

**Figure 4.** The 3DCP-SIT system: (**a**) installation of the SIA on the printing nozzle and (**b**) the upgraded G-code, including commands for both mortar printing and staple insertion.

### 3. Experimental Progress

#### 3.1. Fabrication Process

Typically, there are many interfaces between filaments and layers in the printed sample. To test the inner reinforcing (IR) effect, the rectangular nozzle tip (40 × 10 mm) was installed on the printer. The motion speed of the printing nozzle was 20 mm/s, and the extrusion rate was 6000 mm$^3$/s, which is slightly larger than the required motion speed; subsequently, the extruded paste connected to the previous layer well. The stacked four layers formed a cuboid of 40 × 40 × 200 mm and had only three horizontal interfaces, as shown in Figure 5a–h, referred to as IR-xx samples. Staples were inserted into the previous layer before the next layer was stacked. The staple height was 8 mm, which was less than the cement layer height. Thus, the staples could not penetrate the layer interfaces. The staple patterns included angles of 90° and 45°, spacing distances of 5, 10, and 20 mm, and columns of 1, 2, and 3. To study the interface locking (IL) effect, zigzag-type beam samples for flexural tests were fabricated with the round printing nozzle (diameter 15 mm) and layer thickness of 5 mm, as shown in Figure 5i,j, and named as IL- samples. Staples were inserted over the strip interface and across the layer interface. The average speed of the printing process was 20 mm/s. The extrusion rates for IL-0/1 were 2600 and 2000 mm$^3$/s

for IL-2/3 samples. Thus, IL-0/2 has more cement than IL-1/3. The naming rules for all samples are listed in Table 2. At least three samples were prepared for each type, with failed samples excluded.

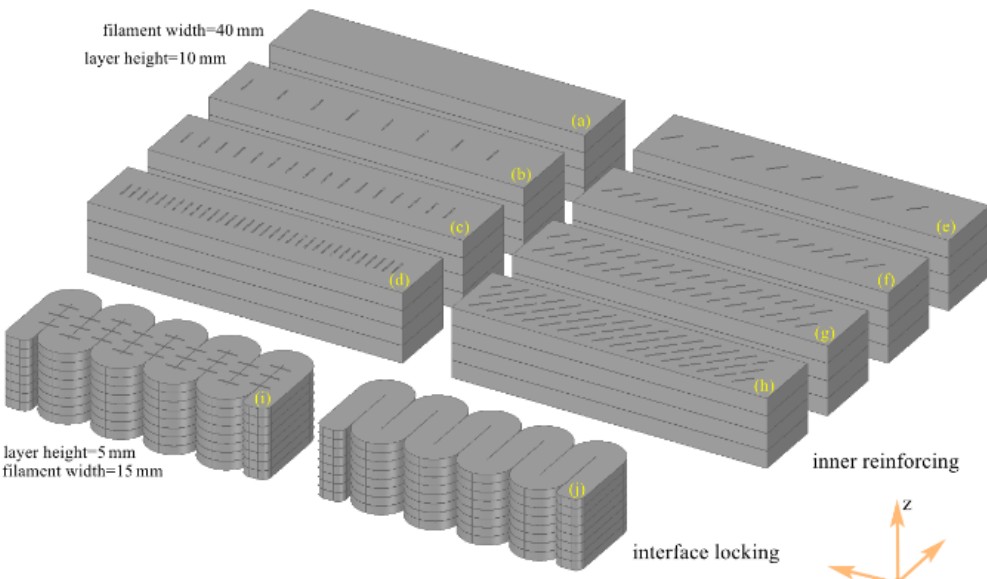

**Figure 5.** Design of the 3DCP-SIA fabricated samples. (**a**) the printed IR- sample without staples, (**b**) the stapled sample with 20 mm spacing and 90°, (**c**) the stapled sample with 10 mm spacing and 90°, (**d**) the stapled sample with 5 mm spacing and 90°, (**e**) the stapled sample with 20 mm spacing and 45°, (**f**) the stapled sample with 10 mm and 45°, (**g**) the stapled sample with 10 mm and 45° in two rows, (**h**) the stapled sample with 10 mm and 45° in three rows, (**i**) the IF- sample with staples, and (**j**) the IF sample without staple.

**Table 2.** Name and parameters of the test samples.

| Name | Figure 5 | Purpose and Nozzle | Filament Width | Filament Height | Angle [1] | Spacing [2] | Staple Column | Density [3] | V% [4] |
|---|---|---|---|---|---|---|---|---|---|
| IR-0 | (a) | | | | | - | | 0 | 0 |
| IR-1 | (b) | | | | 90 | 20 | 1 | 0.125 | 0.24% |
| IR-2 | (c) | Inner Reinforcing with Rectangle Nozzle | 40 | 10 | 90 | 10 | 1 | 0.250 | 0.48% |
| IR-3 | (d) | | | | 90 | 5 | 1 | 0.500 | 0.95% |
| IR-4 | (e) | | | | 45 | 20 | 1 | 0.125 | 0.24% |
| IR-5 | (f) | | | | 45 | 10 | 1 | 0.250 | 0.48% |
| IR-6 | (g) | | | | 45 | 10 | 2 | 0.500 | 0.95% |
| IR-7 | (h) | | | | 45 | 10 | 3 | 0.750 | 1.43% |
| IL-0 | (i) | | | | | - | | | |
| IL-1 | (j) | Interface Locking with Round Nozzle | 15 | 5 | 90 | 10 | | | |
| IL-2 | (i) | | | | | - | | | |
| IL-3 | (j) | | | | 90 | 10 | | | |

1—angle between the staple and the printing direction. 2—spacing, the distance between two adjacent staples. 3—density, the number of staples per cubic centimeter. 4—v%, volume fraction of the staple in the cement.

## 3.2. Fabricating Measurements

The SIA operations were considered successful if the staple body parallels the composite surface and the staple pins vertically into the paste. Obvious nonparallel staples skew to one side when the following composite layer landed, caused large voids, and failed to insert, as shown in Figure 6. The staple density was calculated based on the number of staples per cubic centimeter.

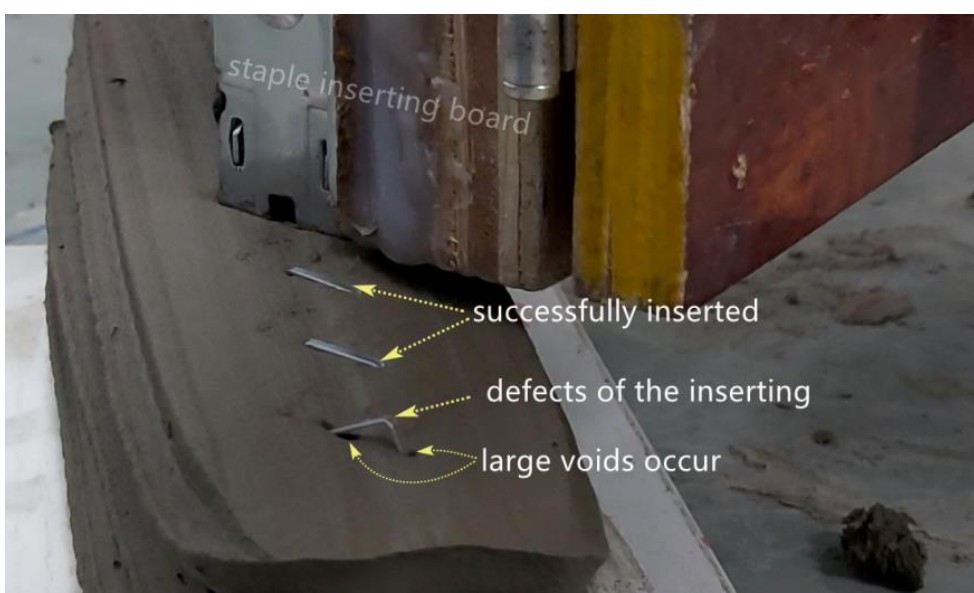

**Figure 6.** Quantification of the staple inserting results.

### 3.3. Tests on Mechanical Properties

All samples were then cured at 100% humidity and room temperature for further tests, as shown in Figure 7, at 28 days. The IR- samples were polished into 40 × 40 × 160 cm cuboids for three-point bending tests at a loading rate of 300 N/s, as shown in Figure 7a. Compressive tests were then conducted on the IR- samples along the Z-axis and a load rate of 300 N/s, as shown in Figure 7b. The IL- samples were subjected to three-point bending tests, as shown in Figure 7c. The failing stresses and strengths were compared and discussed, and the failure modes and contact area between the inserted staples and cement were observed.

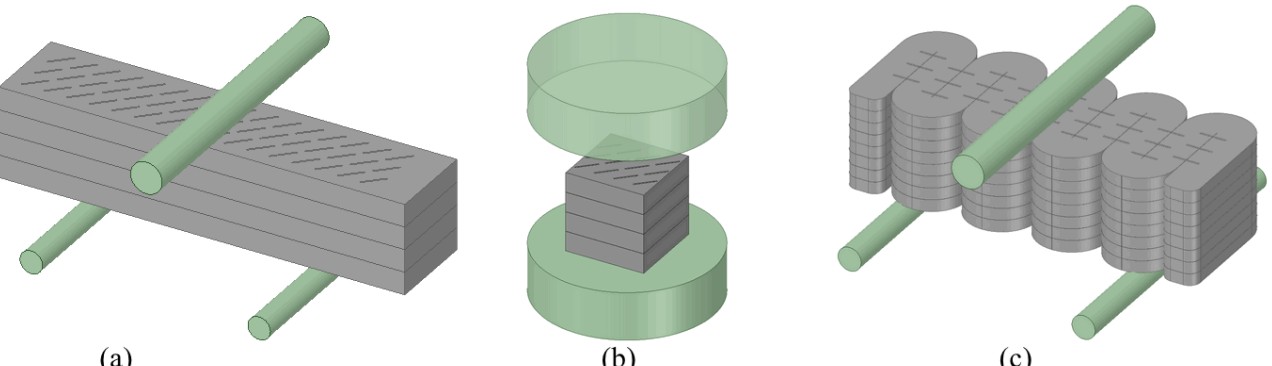

(a)  (b)  (c)

**Figure 7.** Mechanical tests: (**a**) three-point bending for IR- samples along the Z-axis, (**b**) compressive test for IR- samples loading along the Z-axis, and (**c**) three-point bending for IL- samples.

As mentioned in Section 1 and illustrated in Figure 8a, there are interfaces between the layers and strips. The fabricated prisms and flexural loading can be A, B, or C, as shown in Figure 8b, Figure 8c, or Figure 8d, respectively. The IR- samples for flexural and compressive tests belong to type-A and type-E, as in Figure 8b,f, respectively. The IL- samples were for type-B (Figure 8b). Other studies on (Figure 8d) and (Figure 8e) are discussed in later sections.

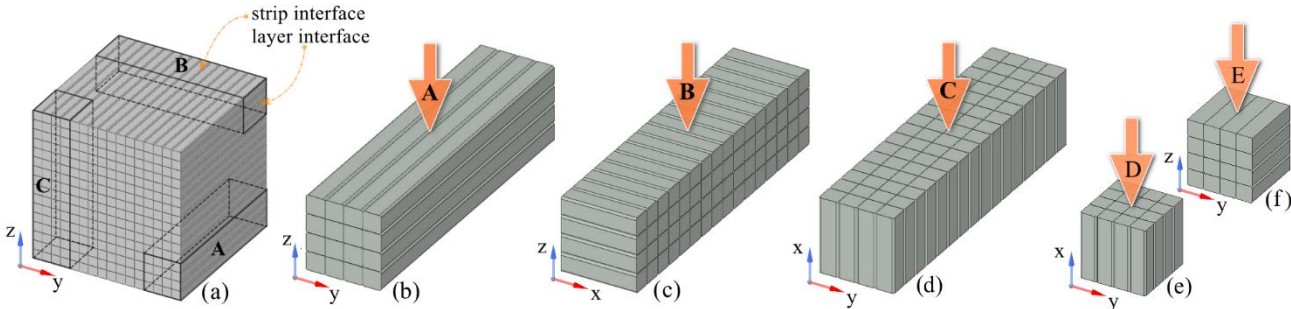

**Figure 8.** Loading directions. (**a**) Positions of the flexural testing samples: (**b**) type-A loaded along the Z-axis, (**c**) type-B loaded along the Z-axis, (**d**) type-C loaded along the X-axis, (**e**) type-D along the X-axis, and (**f**) type-E along the Z-axis.

## 4. Results and Discussion

### 4.1. Fabrication Using 3DCP-SIT

The fabrication processes of all the specimens are shown in Figure 9. The IR- series samples were prepared as shown in Figure 9a–e, with the rectangular nozzle tip, from extruding cement in (a) and (b), switching the nozzle tip to the SIA in (c), starting staple insertion in (d), and finally to finishing the insertion of one layer in (e). The IL- samples were prepared with the SIA and round nozzle tip in turn, as shown in (f) and (g), respectively. The typical samples are shown in (h), (i), and (j), with successfully inserted staples being 98.4% and 98.1% of the staples and two failed staples circled in (h) and (i), respectively. This proved the staple inserting mechanism can be fully automated in bulk manufacturing. The inserting operation extended the fabricating time. Further studies can be conducted on high-speed inserting: staple jetting, which is possible on large-scale printing activities, or developing an inserting mechanism along the material extrusion, as discussed in Section 4.7.

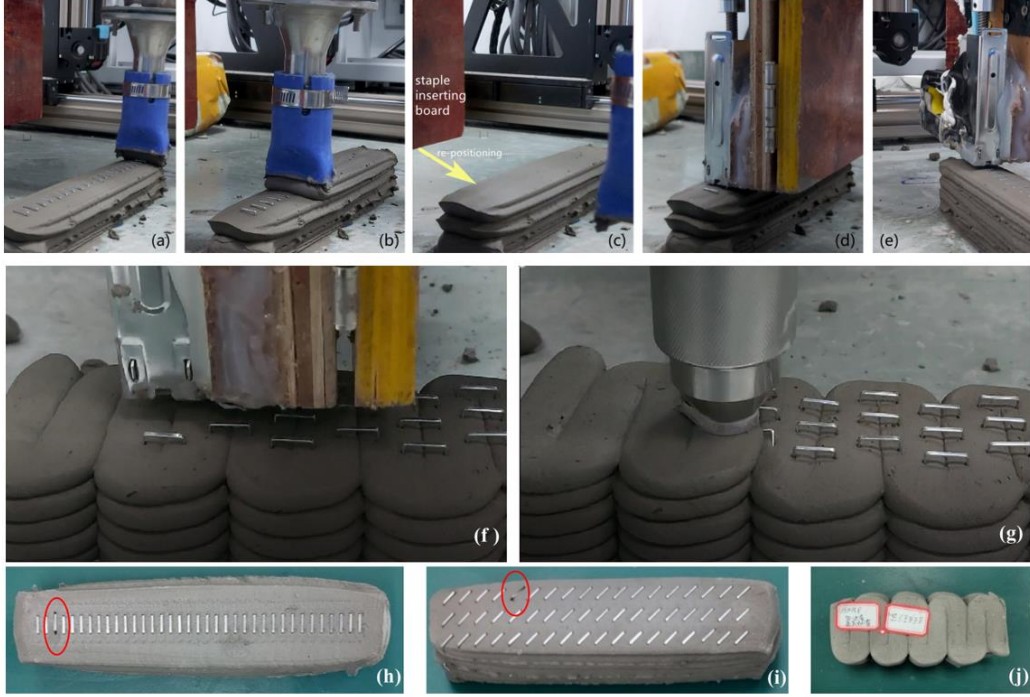

**Figure 9.** Fabricating processes with the 3DCP-SIA and typical samples. (**a**) the mortar extruder was ready to print, (**b**) mortar printing in process, (**c**) SIA repositioning in process, (**d**) staple inserting in process, (**e**) staple insertion was about to finish, (**f**) staple inserting to a IL sample, (**g**) mortar printing to cover the inserted staples, (**h**) a stapled IR sample with a failed staple in red circle, (**i**) a IR sample with three rows of staples, a failed staple in red circle, and (**j**) a printed and stapled IL sample.

### 4.2. Flexural Tests of IR- Samples

Data from the three-point bending tests on the IR- samples are shown in Figure 10a,b. Contrary to expectations, the flexural strengths showed no extensive distinction among all staple inserting methods. All samples failed into two parts immediately after reaching the highest load, with a photograph of a crack shown in Figure 11a, and the stress curves dropped when reaching the maximum value, as shown in Figure 10b.

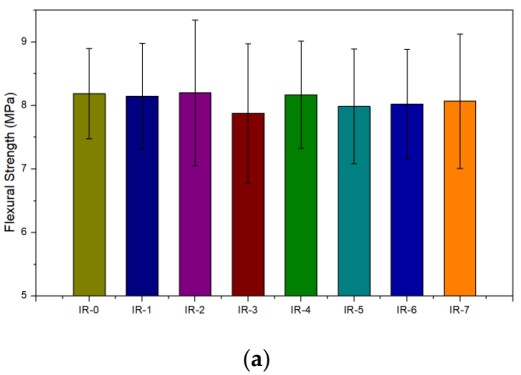

(**a**)

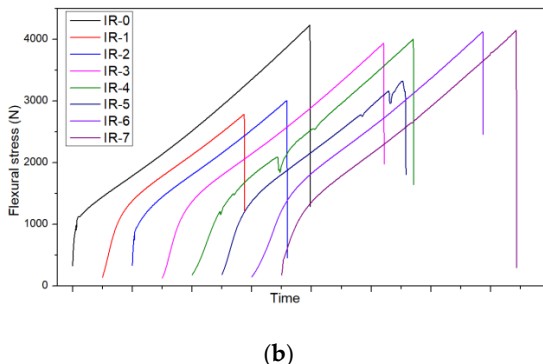

(**b**)

**Figure 10.** Data of the three-point bending tests on the IR- series samples. (**a**) Flexural strengths and (**b**) flexural stresses of typical testing samples, with time axis offsets.

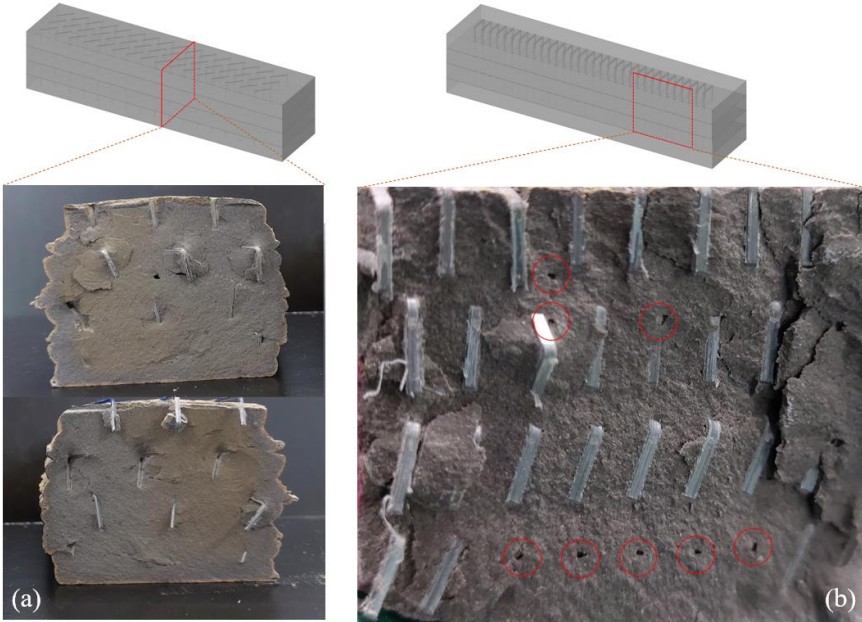

**Figure 11.** Failure of the IR- samples: (**a**) photograph of the broken IR-3 and its location, and (**b**) photograph of the broken IR-7 and its location with gaps in red circles.

The results can explain that the short staples scattered without overlap and functioned separately by locking a narrow cement inside its pin bounds, where there were gaps around the pins in the composite. First, the staple was too short (a length of 11 mm) to reinforce a large amount of composite, while embedded steel cables improved flexural strength by 290%, inserted corrugated fibers improved the strength by 130% [42], and steel mesh achieved 170–290% improvement [43]. Second, in references [14,16–21,44,45], short fibers were mixed before the printing. These mixed fibers were distributed close, near, overlapped, and aligned [46], then functioned together to improve the toughness by 63% [47]. However, in this study, the staples were scattered in the composite and did not overlap much, and thus functioned separately. Third, there were gaps around the staple pins, as shown in Figure 11b, and the bonding was much lower than that of the fibers mixed in the mortar.

Printed outputs with high staple density would require a long time interval between the fabrication of two adjacent layers, causing a decrease of composite interfacial bonding. Further discussion about the inserting equipment is given in Section 4.7.

### 4.3. Compressive Tests on IR- Samples

The compressive test results of the IR- samples are shown in Figure 12a. The inserted staples varied the compressive strength from 16.8 (IR-6) to 22.9 MPa (IR-2) with an average improvement of 5% and maximum improvement of 25%, based on the 18.3 MPa of IR-0. The improvement had an apparent relationship with the staple density (volume fraction), as plotted in Figure 12b. The density of 0.25 staples per cubic centimeter achieved the highest compressive strength of 21.5 MPa, and other densities had no noticeable improvement. This finding indicates that the staple locked the composite under compressive loading and thus increased the compressive strength, which could be referred to as the inner locking effect.

Typically, printed structures have three dimensions of compressive strength anisotropy [48,49]. As in this study, the Z-axis gave the lowest value because the sandwich-like layers were orthogonal to the loading direction. Thus, the testing sample resulted in many cracks and broke into pieces under compressive load. Random cracks still occurred when the IR-samples were reinforced with staples, as proved by the accidental drops in the stress curves, as shown in Figure 13. The central part of the specimen was locked by the staples and remained together as the failure modes shown in Figure 14a–c. The staples contributed to the compressive strength by holding the composite from horizontal extension when under vertical load. To conclude, the inner locking staple could improve the compressive strength along the Z-axis by the inner locking effect.

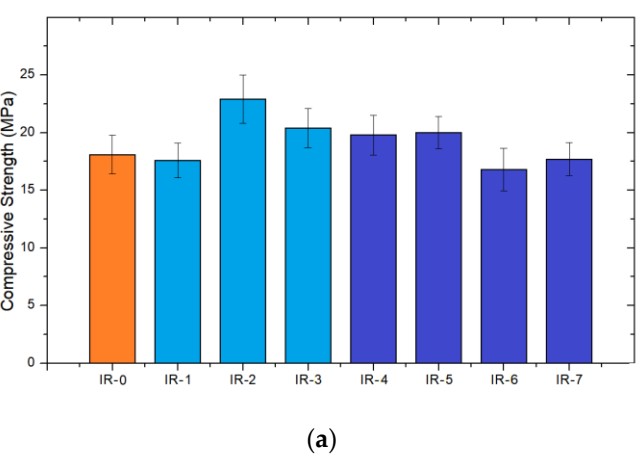

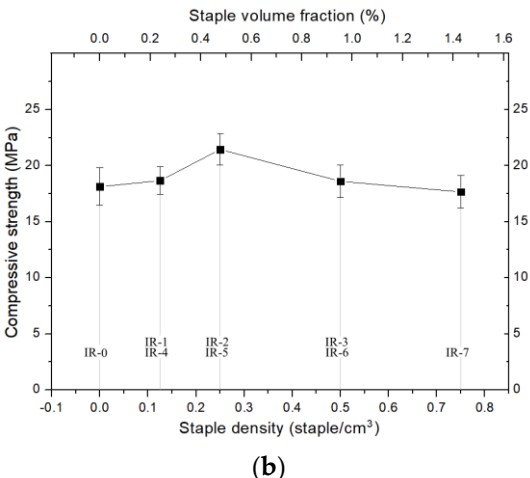

(**a**)         (**b**)

**Figure 12.** Compressive strengths of the IR- samples. (**a**) Strengths vs. sample type and (**b**) strengths vs. staple density/volume fraction.

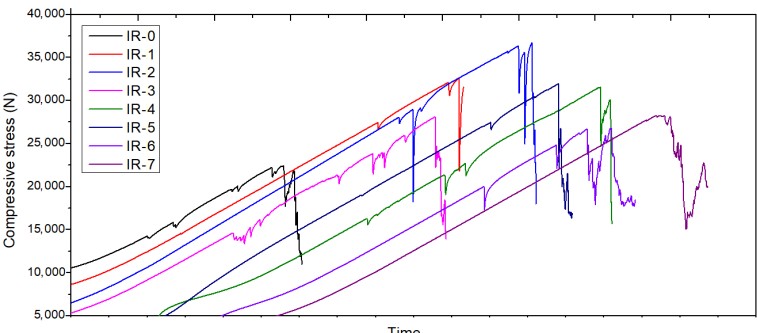

**Figure 13.** Typical compressive stresses of the IR- samples.

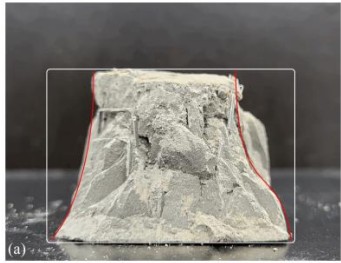
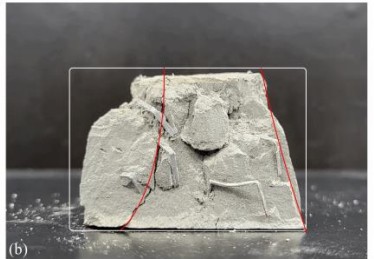
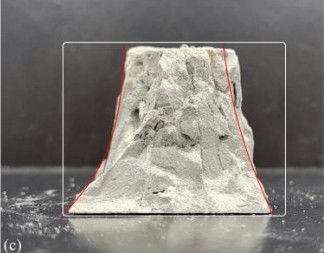

**Figure 14.** Failure photos of the IR- series samples: (**a**) IR-2, (**b**) IR-6, and (**c**) IR-1.

Few researchers have investigated the compressive strength improvement among the studies on the inner reinforcement and across the interfaces. Marchment and Sanjayan [39] conducted compressive experiments vertically to the printing direction, as shown by the *Y*-axis in Figure 8e, and found that the printed sample was 33% lower than the cast samples in the bar-penetrating reinforcement.

For the studies with the same loading direction described herein [Figure 8f], Li and Wang [13] carefully designed the printing paths to spiral-spread the embedded-cables and increased the compressive strength by 50%; Liu and Zhang [32] placed wire meshes between the layers to improve the compressive strength by 75%. The spiraled-cables and placed-meshes enhanced the compressive strength by limiting the material extension. This study achieved a maximum compressive improvement of 18.3% at a volume fraction of 0.48%, as shown in Figure 12a,b. It was lower than the cables and meshes because the short staples functioned separately without overlap, as discussed for flexural results in Section 4.1. Furthermore, poor contact between the staples and composite was the primary defect. The higher the staple density, the more voids, and the mechanical properties declined.

### *4.4. Flexural Tests on IL- Samples*

Three-point bending tests were conducted on the IL- samples, and the bending stresses were then compared instead of the strengths, owing to irregular shape and the same geometry pattern. The central loading was directly onto the middle strip and parallel to the strip interface, as shown in Figure 7c. The average flexural stresses were improved by 46% from 1975 (IL-0) to 2876 N (IL-1), and a 120% improvement from 891 (IL-2) to 1963 N (IL-3). The IL-0/2 samples broke into two pieces with a neat breaking interface, as shown in Figure 15a, and the IL-1/3 samples broke with a rough surface as shown in Figure 15b, proving the staples contributed to the pulling resistance, and softened the post-peak behavior of the samples, as well as in the reference [50].

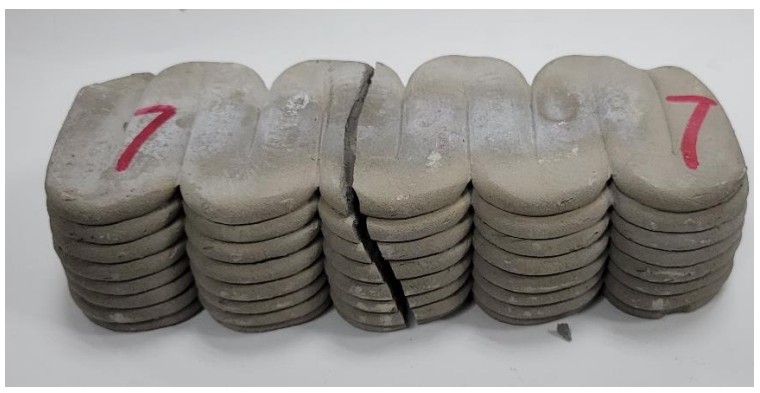

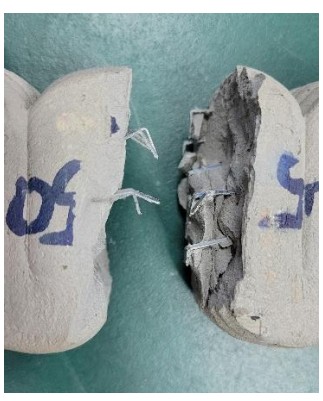

(**a**)                                                     (**b**)

**Figure 15.** Failure of the IL- samples: (**a**) a neat break of IL-0 samples without staples, (**b**) regular breaks of the IL-1/3 samples.

The extruded mortar could be adjusted more (IL-0) or less (IL-2) than the geometry requires by changing the extrusion speed; as mentioned in Section 3.1, the adjacent strips could still join and bond. Compared to IL-0, low material extrusion speed for IL-2 resulted in a lower interface bonding strength since the contact area in the strip interface was less than IL-0 due to less material extruded than in IL-0. Thus, IL-3 showed a relatively significant improvement over IL-2, which was complicated based on the poor interface bonding. We suggest that the 46% improvement in the staple reinforcement of IL-0 was modest.

*4.5. Mechanical Analysis*

This study aimed to investigate the material locking effect between the staple pins. All the staples were inserted vertically to the printed layers, as shown in Figure 5. The flexural and compressive loads were conducted vertically to the printed layers. When a stapled beam was under a bending load (such as a three-point bending test), the inner composite with a staple was under a pulling operation, as shown in Figure 16a. To analyze the mechanism of the staple reinforcement, a simplified model was built, as shown in Figure 16b. There are gaps around the staple pins at positions **a** and **a***'*. Line **b** represents the strip interface. This structure is under a pulling operation. The symbols $f_a$, $f_b$, $f_c$, $f_e$, and $f_f$ represent the maximum failing stresses under pulling loads at positions a, b, c, e, and f, respectively; $f_a$ is determined by the composite strength at positions **a** and **a***'* around the staple pins when the pins are pulled out of the composite; $f_b$ represents the interface bonding stress on the connected strips; $f_c$ is obtained when the staple body fails under the pulling operation; and $f_e$ represents the failure of point e, where the staple pin is a defect of the cement. In addition, $f_f$ corresponds to pure composite failure.

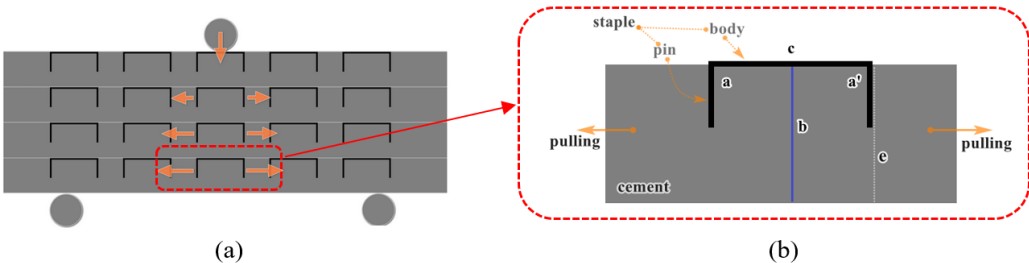

(a)                                  (b)

**Figure 16.** Simple models of the reinforcements. (**a**) Stapled beam under three-point bending test and (**b**) staple over the interface.

If $f_b = 0$ and $f_e = f_f$, the failure stress of this structure under pulling operation is affected by $f_a$ and $f_c$, which are denoted as $f_{ac}$. If $f_a = 0$ and $f_e = f_f$, $f_b$ is the failure stress of the interface bonding. If $f_a \neq 0$, $f_b \neq 0$, and $f_e = f_f$, the maximum pulling stress is $f_{abc}$. Thus, the staple improvement ratio $r$ of the 3DCP-SIT can be calculated using Equation (1):

$$r = \frac{f_{abc} - f_b}{f_b} \times 100\% \tag{1}$$

A low $f_b$ can significantly increase the mechanical improvement value, which is proved by the maximum bending stress improvement of 120% from IL-2 to IL-3, where the interfaces were poorly fabricated by slowing the material extrusion speed compared to IL-0/1. The interfacial bonding could be affected by the printable material proportions and printing parameters [51]. The faster the setting of the material, or the longer the time interval [52], the lower was the bonding of the interface, that is, a low $f_b$.

*4.6. Fabricating Parameters*

There were many gaps around the staples, as shown previously, caused by the insertion process. The voids could not be avoided but could be minimized. Voids also occurred around the penetrated rebar [39] and embedded steel wire [13]. Marchment and Sanjayan observed that the voids were smaller at the deeper point of their penetrated rebar, indicating

better bonding status between the rebar and the composite [39]. Moreover, as shown in Figure 4a, the staples were packed with a film of glue that could further decrease the bonding (the packed steel fibers also have the outer films). The voids and films led to insufficient bonding (i.e., weak contact [13]). The first solution is to minimize the voids by optimizing the inserting parameters or introducing other techniques to the SIA, such as vibrating the staple ultrasonically. The second is to enhance the contact area between the staples and composite, similar to the coating technique. Sun and Gao coated rebar with sand and improved the residual bonding strength by 23% compared to rebar with a smooth surface [33]. The third is the cementitious material. A study [53] showed that the bond quality is highly dependent on the chemical interaction between the cable surface and matrix mortar and the flow behavior of the matrix around the cable. The mortar with high fluidity could fill the gaps, but high fluidity leads to low ability to build. Fortunately, it is possible for large-scale printing with a large nozzle and long loop time between two layers to achieve hardening, planted by large staples. Vibrations and the high-fluidity printable material can minimize the size of the voids.

*4.7. The 3DCP-SIT Technique*

As discussed previously, the staple improvement was lower than those reinforced with meshes, wires, or cables, because the staples were scattered and functioned separately. However, meshes should be cut before putting one printed layer, and embedded wire/cable feeding should sync with the motion speed; otherwise, there will be cable-knot [13] and cable-stretch within the mortar, creating voids and fractures. The staple inserting technique is more straightforward than the mesh/wire/cable reinforcing techniques.

SIA was a prototype in this study. The mortar-printing and staple-inserting were conducted in turns, as shown in Figure 9. Thus, the staple inserting operation extended the time interval between two adjacent layers. We tried to increase the staple density to achieve high staple overlaps. However, the extended time interval weakened the composite bonding between layers. Then, the flexural strengths were not improved either.

Therefore, the 3DCP-SIT system still needs hardware upgrades to increase the insertion speed, reducing the time interval between the printed layers. Furthermore, it can insert staples while the mortar is being extruded, as illustrated in Figure 17, especially for large-scale printing, which needs a large staple size. It is technically feasible that the SIA is the second printing nozzle, and the dual- or multiple-printheads have been successfully utilized in thermoplastic [54], food, or drugs. More techniques could be utilized to achieve further possibilities, such as vibrating the staple, brushing glue on the staple, rotating the insertion angle, or controlling the staple density. The upgraded corresponding software can handle the printing and insertion operations by reading the CAD geometry sketch with staple information, including the locations and directions.

Compared to the meshes/wires/cables, short staples are the short fibers to reinforce printed concrete. Hooked steel fibers can also be used in this inserting technique. Eventually, the 3DCP-SIT fabricates a printed fiber-reinforced concrete structure; the cast steel-fiber reinforced concrete attracted much research attention in past decades and was applied widely in construction [55], including tunneling [56].

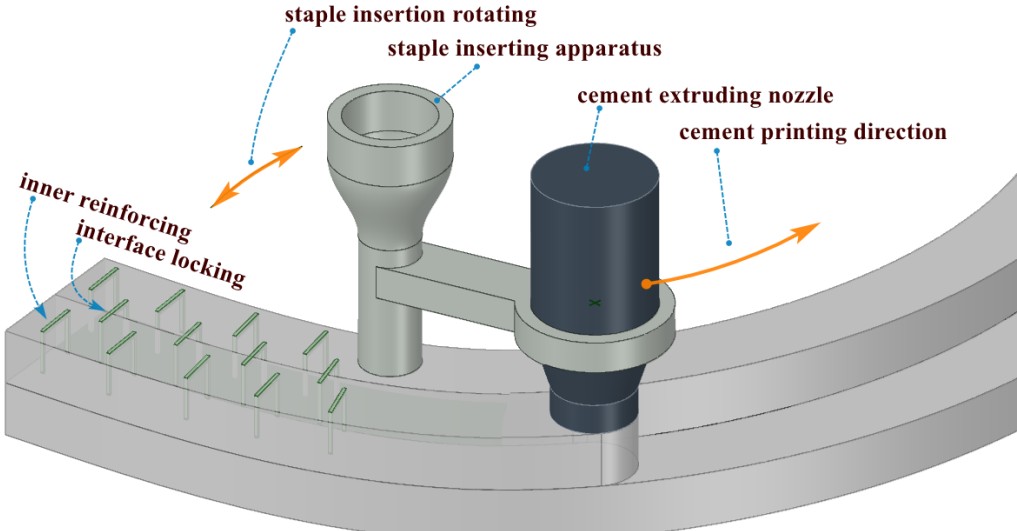

**Figure 17.** Illustration of the 3DCP-SIA system, reinforcing the extruded filaments.

## 5. Conclusions

A staple inserting apparatus was developed and mounted on a 3D printing system, with which 3D-printed and staple-reinforced cementitious specimens were automatically fabricated therein to investigate the mechanical improvement when the staple inserted inner- and inter-strips. The conclusions follow:

- The staples were distributed and aligned as designed entirely using automatic equipment with a success ratio higher than 98%.
- The inner-inserted staples (in IR- samples) showed no noticeable improvement to the flexural strengths because the inserted staples were distributed without overlap and worked alone to lock the composite within the staple pins. They improved the compressive strengths by an average of 5%. Samples with a staple density of 0.25 staples per cubic centimeter had the maximum improvement in compressive strength, which was 25%.
- The inter-inserted staples (in IL- samples) improved the flexural stress by 46–120% by locking the strips. Further analysis showed that the printing parameters can affect the mechanical properties and the improving rate.
- The staple inserting operation introduced voids as defects in the composite properties, and a high staple density decreased the overall mechanical properties. Thus, the inserting technique should be further studied to minimize the voids.
- Future studies should be conducted on reinforcements with high staple density, via an upgraded printing-inserting system which can print cement mortar and insert staples at the same time.

The staples worked alone; thus, the overall improvement of the mechanical properties was not as high as that of long reinforcements such as meshes, cables, and rebar. However, the staple inserting mechanism is simpler than the meshes/wires/cables. Each staple (or inserting operation) is independent of other staples. The 3DCP-SIT system has more prospects in engineering applications. The staples can be considered as fibers to reinforce the printed outputs and have proved certain material locking effects in printed outputs. The proposed 3DCP-SIT system fabricates fiber-reinforced printed concrete structures with controlled fiber distribution by the inserting technique. Further investigations are worth pursuing.

**Author Contributions:** Conceptualization, X.C.; data curation, S.Y.; formal analysis, X.C. and S.Y.; funding acquisition, H.C.; investigation, X.C. and S.Y.; methodology, X.C.; resources, S.Y.; supervision, H.C.; writing—original draft, X.C.; writing—review and editing, H.C. All authors have read and agreed to the published version of the manuscript.

**Funding:** This research was funded by Shenzhen Research Grant (No. KQTD20200909113951005).

**Institutional Review Board Statement:** Not applicable.

**Informed Consent Statement:** Not applicable.

**Data Availability Statement:** Data is contained within the article.

**Acknowledgments:** The work in this study was funded by the Shenzhen Research Grant (No. KQTD20200909113951005).

**Conflicts of Interest:** The authors declare no conflict of interest.

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
