# Peer review of "Experimental Investigation on Inner- and Inter-Strip Reinforcements for 3D Printed Concrete via Automatic Staple Inserting Technique"

_applsci, doi:10.3390/app12042099_

Round 1
Reviewer 1 Report
The proposed idea and its development seem to be of extremely high quality, promising and suitable for practical implementation as a solution to the problem of reinforcing structures in the direct process of building printing.
Comments and recommendations:
- (line 174, 175) Please explain why staples with a higher height were not used, which can penetrate the boundary between the layers and connect them. Technically, construction staplers generally allow staples up to 16mm high. This would make it possible to ensure the interaction of reinforcing clips from different layers, and, thus, would allow obtaining continuous reinforcement of the composite along the height. This is carried out with other traditional methods of reinforcement with which a comparison is made in the article.
- (line 233) Indicated by the authors - insufficient increase in tensile strength of samples IR-4, 5, 6, 7 is associated with an unsuitable scheme of replacement of reinforcing staples. As the authors note, when using the chosen scheme, well-reinforced sections are formed inside the row of staples and an unreinforced gap between the rows, as well. The main problem is the occurrence of a weakened area, which is formed by the legs (pins) of a large number of staples. Destruction occurs in this area. Please explain why, with a large amount of high-quality work done, such an imperfect scheme for reinforcing elements replacing was chosen, contrary to existing canons, which does not allow using the strength potential of reinforcement? It would probably be much more efficient to replace the staples in a checkerboard pattern or with varying offsets between rows.
- (line 240) The lack of increase in tensile strength due to the formation of gaps (defects, voids) around the legs of the staples during their introduction is not convincing, since the subsequent stacked layer of the mixture deforms the underlying one. This contributes to the filling of possible defects and improves the anchoring of the reinforcing elements. The destruction along the surface in which the legs of the staples are located is associated with their unsuitable placement (see Comment 2).
- Please, indicate how many identical samples were tested for determining the values of mechanical parameters (tensile and compressive strength).
Taking into account these comments, it is recommended to further emphasize that the main goal was precisely the development of a method and equipment for the introduction of reinforcing staples, as well as its laboratory testing in various modes.
It might be worth reflecting this idea in the following:
- in the correction of the title of the article;
- some reduction of section 4, for example by deleting section 4.5, which, taking into account the comments made, seems to be uninformative;
- bring the most promising schemes of placement of staples as planned for further research.
The above recommendations will mitigate the main comments on the work and further increase its scientific and practical value.
Author Response
Dear Reviewer,
Please see the attached file. Many thanks.
With best regards,
The Authors

Reviewer 2 Report
Dear authors,
I found the topic of your paper really interesting. The article was well written and presents a sufficient scientific methodology. I suggest you to add some references suitable for your topic:
- Lo Giudice A, Ronsivalle V, Grippaudo C, et al. One Step before 3D Printing-Evaluation of Imaging Software Accuracy for 3-Dimensional Analysis of the Mandible: A Comparative Study Using a Surface-to-Surface Matching Technique. Materials (Basel). 2020;13(12):2798. Published 2020 Jun 21. doi:10.3390/ma13122798
- Matarese G, Isola G, Ramaglia L, Dalessandri D, Lucchese A, Alibrandi A, Fabiano F, Cordasco G. Periodontal biotype: characteristic, prevalence and dimensions related to dental malocclusion. Minerva Stomatol. 2016 Aug;65(4):231-8. Epub 2016 Apr 1. PMID: 27035270
Good work.
Author Response

(The authors gave the same response as above.)

Reviewer 3 Report
The introduction needs to highlight the importance of the study and it would be good to make relevance to the main industry of interest. It is suggested to refer to the following: https://doi.org/10.1680/jstbu.18.00136.
The sentences in the Introduction seem to be distorted, for example, after reference [21], the sentence doesn’t seem to be in the right place and does not connect with the previous paragraph.
In the introduction, it would be good to refer to similar studies and make a comparison: especially shape retention, mix formulation for low carbon cementitious composites and fibre reinforcements: https://doi.org/10.1016/j.conbuildmat.2020.118928; https://doi.org/10.1016/j.matdes.2021.109574; https://doi.org/10.1007/s13204-021-01738-2
The aims and objectives of the study need to be better presented. It would be good to re-write it and make it clearer what the novelty of the study is and how the authors address the challenges presented in the manuscript.
It would be better to justify why all the analytical and bulk property tests were carried out in the methodology section?
The results in section 4.1 don't seem to have any discussion or purpose. The narrative needs improvement.
The flexural strength is low and it is clear that the stapling has not had any influence on improving it. But that in itself is useful data for other researchers. Perhaps the length of the staple should be increased. Authors must comment on it based on their observations.
The inter-layer staples improved the strength but it would have been good to have some microstructural analysis of the interfacial zones.
The conclusions can be written concisely with more facts presented from the study in terms of numbers and percentage improvements.
Author Response

(The authors gave the same response as above.)

Reviewer 4 Report
First of all, I congratulate the authors for their work.
In Fig. 4.b the nailing there appears G-Code. I suggest one final submission to be added as well the files. Complete files.
Author Response

(The authors gave the same response as above.)

Round 2
Reviewer 1 Report
Thank you, all comments are taken into account
Reviewer 3 Report
Can now be accepted.